

# Relationship between triterpenoid anticancer drug resistance, autophagy, and caspase-1 in adult T-cell leukemia

Tsukasa Nakanishi[1,2], Yuan Song[1,3,*], Cuiying He[1,*], Duo Wang[1,*],
Kentaro Morita[1], Junichi Tsukada[2], Tamotsu Kanazawa[1] and
Yasuhiro Yoshida[1]

[1] Department of Immunology and Parasitology, School of Medicine, University of Occupational and Environmental Health, Kitakyushu, Japan
[2] Department of Hematology, University of Occupational and Environmental Health, Kitakyushu, Japan
[3] Department of Clinical Laboratory, Fourth Hospital of Hebei Medical University, Shijiazhuang, China
* These authors contributed equally to this work.

## ABSTRACT

We previously reported that the inflammasome inhibitor cucurbitacin D (CuD) induces apoptosis in human leukemia cell lines. Here, we investigated the effects of CuD and a B-cell lymphoma extra-large (Bcl-xL) inhibitor on autophagy in peripheral blood lymphocytes (PBL) isolated from adult T-cell leukemia (ATL) patients. CuD induced PBL cell death in patients but not in healthy donors. This effect was not significantly inhibited by treatment with rapamycin or 3-methyladenine (3-MA). The Bcl-xL inhibitor Z36 induced death in primary cells from ATL patients including that induced by CuD treatment, effects that were partly inhibited by 3-MA. Similarly, cell death induced by the steroid prednisolone was enhanced in the presence of Z36. A western blot analysis revealed that Z36 also promoted CuD-induced poly(ADP ribose) polymerase cleavage. Interestingly, the effects of CuD and Z36 were attenuated in primary ATL patient cells obtained upon recurrence after umbilical cord blood transplantation, as compared to those obtained before chemotherapy. Furthermore, cells from this patient expressed a high level of caspase-1, and treatment with caspase-1 inhibitor-enhanced CuD-induced cell death. Taken together, these results suggest that rescue from resistance to steroid drugs can enhance chemotherapy, and that caspase-1 is a good marker for drug resistance in ATL patients.

## INTRODUCTION

Adult T-cell leukemia (ATL) is an aggressive and malignant blood disease. There are many ATL patients in the Kyushu area of Japan who have a very poor prognosis even with intensive chemotherapy (*Iwanaga, Watanabe & Yamaguchi, 2012*), owing to rapid disease progression and frequent infectious complications. The median survival of patients in Japan is six months (*Bazarbachi et al., 2004*). Several drugs including synthetic

Corresponding author
Yasuhiro Yoshida,
freude@med.uoeh-u.ac.jp

glucocorticoids are used in chemotherapy; however, the treatment is not completely effective due to the development of drug resistance.

Cucurbitacin (Cu) is a type of cucurbit belonging to the triterpenoid group of molecules with a steroid structure (Fig. 1A). We previously reported that CuD has antitumorigenic activity (*Takahashi et al., 2009*) and showed that it acts as a proteasome inhibitor and induces apoptosis in leukemia cell lines with differential sensitivity (*Ding et al., 2011*). Glucocorticoid steroid hormones are universally used in the treatment of acute lymphoblastic leukemia, but the development of resistance is associated with poor prognosis (*Den Boer et al., 2003*). However, the mechanistic basis for the development of resistance to steroid-type molecules by ATL cells is not well understood.

Autophagy is an evolutionarily conserved mechanism for the degradation of cellular waste components and is critical for the maintenance of homeostasis (*Puleston & Simon, 2014*). Defective autophagy is associated with diseases such as cancer; it has been implicated in tumor development and survival in blood diseases (*Bosnjak et al., 2014*; *Nencioni et al., 2013*; *Pierdominici et al., 2014*). Some studies have reported the induction of cell death via inhibition of autophagy in treatments against T-cell leukemia (*Jia et al., 1997*; *Nahimana et al., 2009*; *Wang et al., 2014*), but there are no similar reports for ATL, and the extent to which autophagy is involved in the induction in chemically induced cell death in ATL is unclear.

During the preparation of this communication, *Paugh et al. (2015)* reported caspase-1-related glucocorticoid resistance in leukemia cells (*Paugh et al., 2015*). In this study, we investigated the antitumorigenic effects of another steroid-structured agent, CuD, in primary ATL cells with respect to autophagy involvement and drug resistance.

## MATERIALS AND METHODS

### Reagents and antibodies

CuD was purchased from Nacalai Tesque (Kyoto, Japan) and dissolved in 10% ethanol to a stock concentration of 1 mg/mL. Z36 was obtained from Enzo Life Sciences (Farmingdale, NY). The autophagy inhibitor 3-MA was purchased from Sigma (St. Louis, MO, USA) and was diluted in phosphate-buffered saline. The anti-poly ADP ribose polymerase (PARP) antibody (9542S) was purchased from Cell Signaling Technology (Danvers, MA, USA). The $\beta$-actin (A5441) antibody was obtained from Sigma; the anti-microtubule-associated protein 1A/1B-light chain 3 (LC3) antibody (M186-3) was purchased from MBL (Nagoya, Japan). Antibodies against α-tubulin (H-300; sc-5546) and caspase-1 (M-20; sc-514) were obtained from Santa Cruz Biotechnology (Santa Cruz, CA, USA). Caspase-1 inhibitor (#400010) was obtained from Calbiochem (San Diego, CA, USA).

### Preparation of human peripheral blood lymphocytes (PBLs)

PBLs were obtained from ATL patients or healthy donors following the ethical guidelines of University Occupational and Environmental Health, Japan (No. H23-061); donor information is summarized in Table 1. Cells were harvested three times from ATL patient number 1: once before chemotherapy; after three rounds of THP-COP chemotherapy

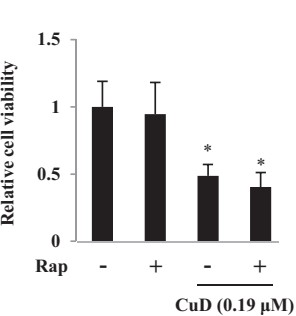

**A**

cucurbitacin D                    prednisolone

**B   Healthy donor**        **C   ATL Patient**

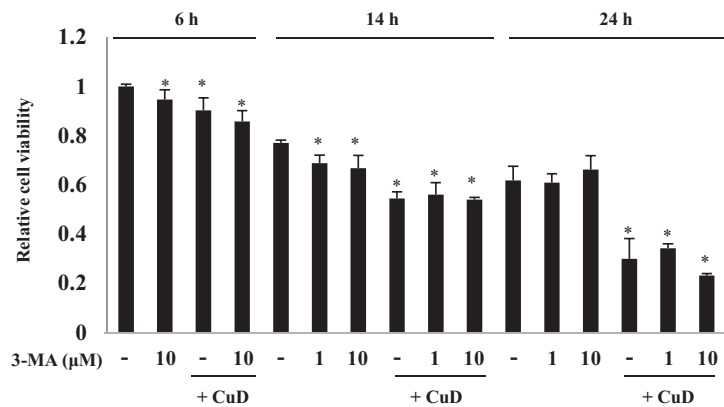

**D     ATL Patient**

**Figure 1 CuD induces primary ATL patient cell death.** (A) Structures of CuD and prednisolone. (B, C) PBLs were freshly prepared from healthy donor (B) or ATL patients (C). Cells ($1 \times 10^4$/well) were treated with indicated concentrations of CuD or vehicle in the presence or absence of rapamycin (100 $\mu$M) for 24 h. (D) PBLs ($1 \times 10^4$/well) from ATL patient were treated with CuD (0.19 $\mu$M) in the presence or absence of 3-MA (10 $\mu$M) for indicated periods. Cell viability was assessed with the Cell Titer-Glo assay. Results are expressed as mean activity relative to controls and SD from quadruplicate cultures. $^*P < 0.05$ vs. vehicle without CuD treatment at each time point.

[30 mg/m$^2$ pirarubicin (THP-ADR) for 1 day, 600 mg/m$^2$ cyclophosphamide for 1 day, 1 mg/m$^2$ vincristine for 1 day, and 30 mg/m$^2$ prednisolone for 5 days] but before umbilical cord blood transplantation; and at the time of recurrence 96 days after the transplantation. Cells were isolated using Lymphoprep (Axis Shield, Oslo, Norway)

**Table 1 Characteristics of ATL patients and healthy donors.** PBLs were obtained from ATL patients or healthy donors as following ethics in University of Occupational and Environmental Health, Japan. Donor information is summarized in Table 1.

| | Disease | Sex | Age (years) | HTLV-1 | Tax | Lymphoid cells (%) | CuD treatment (inhibition to Ct, %) | CuD + Z36 treatment (inhibition to Ct, %) |
|---|---|---|---|---|---|---|---|---|
| Healthy donor 1 | None | M | 29 | – | – | | | |
| Healthy donor 2 | None | F | 31 | – | – | | | |
| Healthy donor 3 | None | F | 27 | – | – | | | |
| Healthy donor 4 | None | F | 37 | – | – | | | |
| Patient 1 | Acute ATL Before chemo | M | 68 | + | + | 14 | 54 | 71 |
| | Acute ATL After chemo | | | | | n.e. | 2 | 0 | 8 |
| | Acute ATL Recurrence after umbilical cord blood transplantation | | | | | n.e. | 14 | 2 | 43 |
| Patient 2 | Acute ATL | M | 61 | + | + | 65 | 24 | 62 |
| Patient 3 | Acute ATL | F | 75 | + | + | 8 | 38 | 60 |
| Patient 4 | Acute ATL | F | 75 | + | + | 11 | 7 | 52 |
| Patient 5 | Acute ATL | M | 64 | + | n.e. | 2 | 10 | 22 |
| Patient 6 | Acute ATL | M | 82 | + | n.e. | 28 | 16 | 46 |
| Patient 7 | Chronic ATL | M | 52 | + | + | 41 | 55 | 94 |
| Patient 8 | Mycosis fungoides | M | 63 | – | – | 14 | 34 | 70 |
| Patient 9 | T-cell lymphoma | M | 63 | – | – | 7 | 0 | 92 |
| Patient 10 | Acute ATL | F | 65 | + | n.e. | 94 | 15 | 66 |

**Note:**

M, male; F, female.

according to the manufacturer's protocol. Briefly, 6 ml of blood were overlaid on 3 ml of Lymphoprep reagent and centrifuged at $600 \times g$ for 30 min. After removing the PBL layer, cells were washed with PBS, resuspended in medium, and counted. Prepared cells were cultured as described below.

## Proliferation assay

Cell proliferation was assessed with the Cell Titer-Glo luminescent cell viability assay (Promega, Madison, WI, USA) as previously described (*Ding et al., 2011*). Briefly, cells were seeded in 96-well plates at $1 \times 10^4$/well, and then treated with Z36 and/or CuD in the presence or absence of 3-MA. A 10-$\mu$l volume of assay reagent was added to each well, and fluorescence was measured with a luminometer (Luminescencer-JNR-II; ATTO, Tokyo, Japan).

## Western blotting

Cells were lysed with radio immunoprecipitation assay buffer (RIPA) (*Yoshida et al., 2009*) to obtain whole-cell extracts. Equivalent amounts of protein (10 $\mu$g) were resolved by sodium dodecyl sulfate-polyacrylamide gel electrophoresis, then transferred to and

immobilized on nitrocellulose membranes (Amersham, Buckinghamshire, UK) that were probed with the appropriate primary and secondary antibodies. Protein bands were detected using the ImmunoStar LD detection system (Wako, Osaka, Japan), and the signal intensity was quantified using ImageJ software (National Institutes of Health, Bethesda, MD, USA). Expression levels of target proteins were normalized to that of $\beta$-actin or $\alpha$-tubulin.

### Statistical analysis

Results are expressed as mean ± SD and inter-group differences were evaluated by analysis of variance with Scheffe's post-hoc analysis. A $P < 0.05$ was considered statistically significant.

## RESULTS

### CuD induces death in primary cells from ATL patients

We examined the effects of CuD on primary ATL patient cells. The cell viability assay confirmed that CuD was not toxic to PBLs from healthy donors (Fig. 1B), as we have previously reported (*Ding et al., 2011*). Activated PBL cells by mitogen were also unaffected by CuD (Fig. S1A). In contrast, CuD markedly reduced the viability of primary ATL patient cells (Fig. 1C). To examine the involvement of autophagy, we treated cells with rapamycin, a mammalian target of rapamycin inhibitor and autophagy inducer, and found that it had no effect on CuD-induced cell death (Fig. 1C). Similarly, treatment with the autophagy inhibitor 3-MA did not affect CuD-induced antitumorigenic activity in primary ATL patient cells, while 3-MA treatment alone slightly decreased cell viability (Fig. 1D).

### Z36 inhibits proliferation and enhances CuD-induced cell death in primary ATL patient cells

The effects of the B-cell lymphoma (Bcl)-2 inhibitor Z36 on ATL patient cells before chemotherapy were examined. Low concentrations (3 and 5 $\mu$M) of Z36 were not toxic to PBLs from healthy donors; however, cell viability was decreased at a high concentration (7.5 $\mu$M) (Figs. 2A and S2B). Application of Z36 inhibited the proliferation of ATL patient cells in a dose-dependent manner (Fig. 2B, from Patient 1; other data shown in Fig. S2A) and enhanced CuD-induced cell death, which was partly abrogated by treatment with 3-MA (Figs. 2C and S3). These results were confirmed in 10 patients and three healthy controls, although only a representative case is shown. Z36 had similar effects on prednisolone-treated ATL patient cells (Fig. 2D), suggesting that cell death induced by steroid-like agents acts via a different mechanism than Z36-induced cell death.

### Autophagy negatively regulates CuD- or Z36-induced PARP cleavage

We examined the levels of cleaved PARP and LC3-II proteins by western blotting, and found that Z36 induced PARP cleavage in primary ATL patient cells, contrary to previous reports (*Lin et al., 2009*), while Z36 treatment increased LC3-II expression in these cells, as previously reported (Fig. 3A). Treatment with the autophagy inhibitor 3-MA

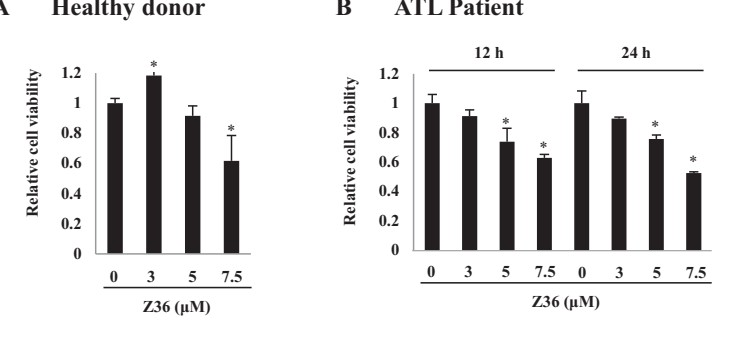

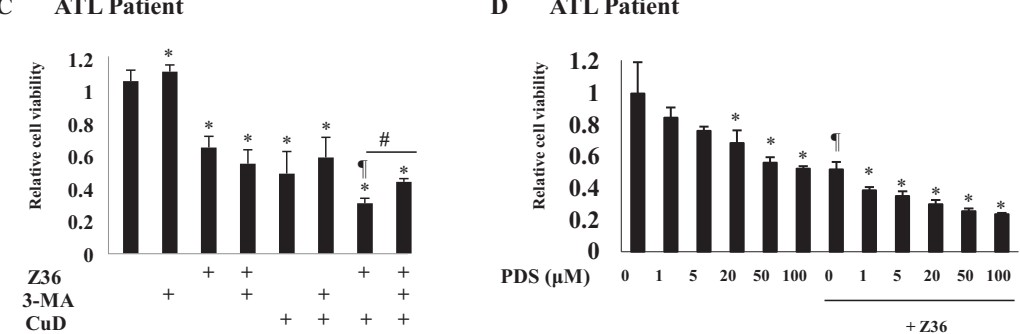

**Figure 2 Z36 inhibits proliferation and enhances CuD-induced cell death in primary ATL patient cells.** (A–D) PBLs ($1 \times 10^4$/well) from healthy donors (A) or ATL patients (B–D) were treated with indicated concentrations of Z36 or dimethyl sulfoxide (DMSO, a control solvent for Z36) for 12 or 24 h, or CuD (0.19 μM) or prednisolone for 24 h. (C) PBLs from ATL patients were pre-treated with 10 mM 3-MA or vehicle for 2 h, and then exposed to 0.19 μM CuD, Z36 (7.5 μM), or vehicle (Et-OH, 0.01%) for 24 h. (D) Cell viability was determined using the Cell Titer-Glo assay. Results are expressed as mean activity relative to controls and SD from quadruplicate cultures. *$P < 0.05$ vs. vehicle; ¶$P < 0.05$ vs. treatment with CuD only; #$P < 0.05$ vs. CuD and Z36 co-treatment.

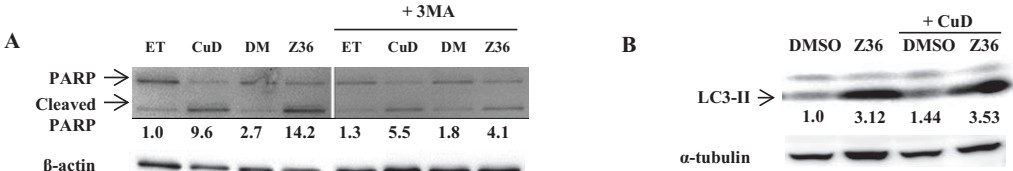

**Figure 3 Autophagy negatively regulates CuD- or Z36-induced PARP cleavage in ATL patient cells.** (A, B) PBLs ($2 \times 10^5$/well) from ATL patients were treated with CuD (0.19 μM), Z36 (7.5 μM), or vehicle in the presence or absence of 3-MA (10 mM) for 24 h. Whole cell extracts were used for western blot analysis of cleaved PARP (A) and LC3-II (B) expression; *β*-actin or tubulin was used as a loading control. ET, ethanol; DM, DMSO. Western blotting densitometry data were quantified and are presented below their images after calculating by loading contents levels.

reduced levels of cleaved PARP proteins induced by Z36 and CuD (Fig. 3A). Co-treatment with CuD and Z36 did not increase LC3-II expression relative to cells treated with Z36 only (Fig. 3B).

**A    ATL Patient after chemo**

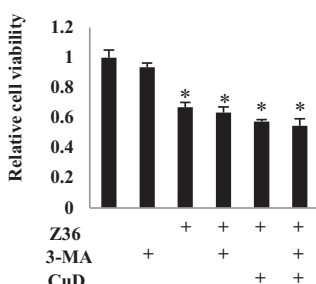

**C    Recurrence after the umbilical cord blood transplant**

**B**

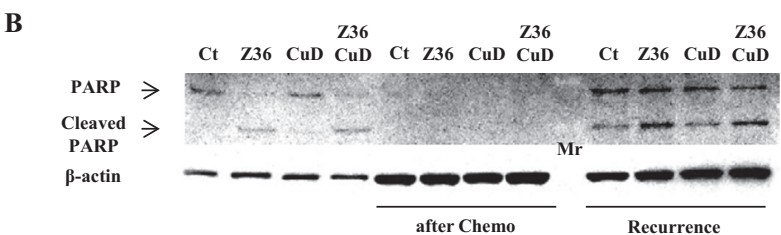

**D**

**E**

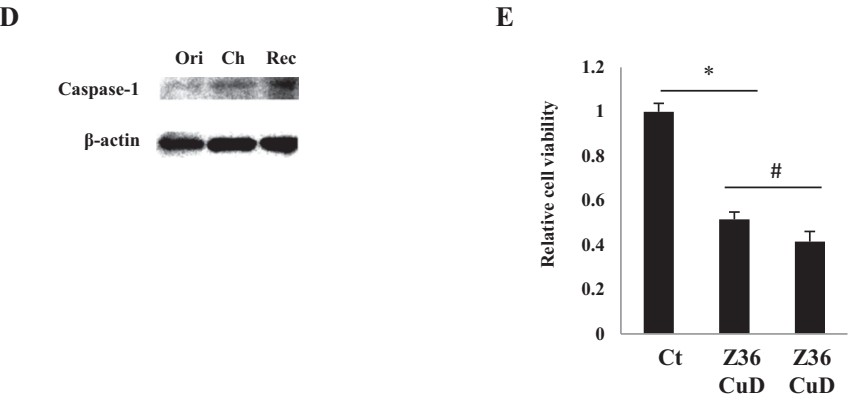

**Figure 4 CuD and Z36 show limited effects on ATL patient-derived primary cells upon disease recurrence.** (A, C) PBLs ($1 \times 10^4$/well) from ATL patients after chemotherapy (A) or from ATL patients upon recurrence after umbilical cord blood transplantation (C) were treated with Z36 (7.5 µM) or vehicle for 24 h. Cell viability was determined with the Cell Titer-Glo assay. Results are expressed as mean activity relative to controls and SD from quadruplicate cultures. (B) PBLs ($1 \times 10^4$/well) from a patient before therapy (left four lanes), after umbilical cord blood transplantation (middle four lanes), and after recurrence (right four lanes) were treated with CuD (0.19 µM) and/or Z36 (7.5 µM) or with vehicle for 24 h. Whole cell extracts were used for western blot analysis of cleaved PARP; $\beta$-actin was used as a loading control. Mr, molecular marker. (D) Whole cell extracts from a patient before therapy (ori), after umbilical cord blood transplantation (Ch), and after recurrence (Rec) were analyzed for caspase-1 expression by western blotting; $\beta$-actin was used as a control. (E) ATL patient cells ($1 \times 10^4$/well) obtained upon recurrence after umbilical cord blood transplantation were treated with CuD (0.19 µM) and Z36 (7.5 µM) or vehicle for 24 h in the presence or absence of caspase-1 inhibitor (20 µM). Cell viability was determined with the Cell Titer-Glo assay. Results are expressed as mean activity relative to controls and SD from quadruplicate cultures. *$P < 0.05$ vs. vehicle; #$P < 0.05$ vs. CuD and Z36 co-treatment.

## CuD and Z36 have limited effects in primary ATL patient cells after disease recurrence

As expected, treatment with CuD (0.19 μM) and/or Z36 (7.5 μM) and/or 3-MA (10 mM) had no effect on primary ATL patient cell viability after chemotherapy, since there were no leukemia cells (Fig. 4A). Furthermore, PARP cleavage was not induced by CuD and/or Z36 in these cells after chemotherapy (Fig. 4B, middle four lanes). Interestingly, the effects of CuD and Z36 were attenuated in primary ATL patient cells upon recurrence after umbilical cord blood transplantation (Fig. 4B, right four lanes) as compared to the post-chemotherapy state (Figs. 4B and 4C). Increased caspase-1 levels have been linked to glucocorticoid resistance in leukemia cells (*Paugh et al., 2015*). We found here that caspase-1 levels were increased after chemotherapy, implying the development of drug resistance. Consistent with this supposition, the effects of combined CuD and Z36 treatment on primary ATL patient cells were enhanced in the presence of a caspase-1 inhibitor (Fig. 4E).

## DISCUSSION

There is growing interest in natural products as a source of new drugs for cancer therapy. We previously reported that one such product, CuD, could induce human T-cell leukemia cell death in vitro via proteasome inhibition (*Ding et al., 2011*). In the present study, we demonstrated that CuD inhibits the proliferation of primary ATL patient cells. This is the first evidence of an inhibitory effect by a member of Cucurbitaceae on tumor cells. On the other hand, CuD showed no toxicity to peripheral blood mononuclear cells from healthy donors or primary cells from mouse (*Song et al., 2013*), and it was in fact found to enhance PBL proliferation in the current study. Conversely, CuD strongly induced primary ATL patient cell death, as determined by the cell viability assay and by direct counting following CuD treatment (Fig. S4). As expected, primary ATL patient cells after chemotherapy were unaffected by CuD or Z36 treatment, but cells obtained upon recurrence showed reduced responsiveness to CuD and Z36 after umbilical cord transplantation as compared to those obtained from the first occurrence of the disease. Given that recurrence after transplantation is associated with poor prognosis (*Hishizawa et al., 2010*), our results suggest that these cells developed drug resistance. Caspase-1 level is a marker for glucocorticoid (e.g., prednisolone) resistance in leukemia cells (*Paugh et al., 2015*); indeed, we observed an upregulation of caspase-1 expression in primary ATL patient cells after recurrence. However, caspase-1 level was unexpectedly elevated after transplantation, suggesting that drug resistance was induced at this stage.

Given that drug resistance influences chemotherapeutic trials, developing novel treatment strategies and identifying new target molecules represent an ongoing challenge. The role of autophagy in hematological malignancies has been widely investigated (*Macintosh & Ryan, 2013*). Autophagy inhibitors have been shown to overcome cell resistance to anti-cancer drugs (*Carew et al., 2007*); autophagy induction is required to overcome glucocorticoid resistance in acute lymphoblastic leukemia cells (*Bonapace et al., 2010*). Bortezomib is a proteasome inhibitor and apoptosis inducer

(*Pei, Dai & Grant, 2003*) that is used clinically for the treatment of multiple myeloma (*Sonneveld et al., 2012*). It was previously shown that bortezomib treatment in the presence of autophagy inhibitors enhances apoptosis (*Hoang et al., 2009*; *Jia et al., 2012*). This finding is consistent with our observation that the effects of the proteasome inhibitor CuD were enhanced in the presence of 3-MA. On the other hand, autophagy has been shown to enhance apoptosis induction in blood malignancies (*Fang et al., 2012*), which is similar to our results obtained by co-treatment with Z36, a Bcl-xL inhibitor and autophagy-specific inducer in HeLa cells or THP-1 cells (*Casalino-Matsuda et al., 2015*). Indeed, we found that Z36 treatment increased primary ATL patient cell death by inducing not only LC3-II expression but also PARP cleavage, suggesting that it functions as an apoptosis inducer in ATL cells, which has not been previously reported. Z36 was only found to suppress PBL proliferation at a concentration of 3 $\mu$M or greater, implying that lymphoid cells are more sensitive to Z36 than healthy donor cells. In this scenario, autophagy protects against chemically induced apoptosis.

In conclusion, we described the distinct effects of two Bcl-xL inhibitors, which were abolished in the presence of a caspase-1 inhibitor. Based on these findings, acquired drug resistance should be considered in the context of the type of cell death that is induced by the levels of specific molecules such as caspase-1.

### Funding
This study was supported in part by a Grant-in-Aid for Scientific Research (A) (no. 25241015) and a Grant-in-Aid for Scientific Research (B) (no. 24390159) from the Japan Society for the Promotion of Science to Y. Yoshida. The funders had no role in study design, data collection and analysis, decision to publish, or preparation of the manuscript.

### Grant Disclosures
The following grant information was disclosed by the authors:
Grant-in-Aid for Scientific Research (A): 25241015.
Grant-in-Aid for Scientific Research (B): 24390159.

### Competing Interests
The authors declare that they have no competing interests.

### Author Contributions
- Tsukasa Nakanishi performed the experiments, analyzed the data, contributed reagents/materials/analysis tools, wrote the paper, prepared figures and/or tables.
- Yuan Song performed the experiments.
- Cuiying He performed the experiments.
- Duo Wang performed the experiments.

- Kentaro Morita performed the experiments.
- Junichi Tsukada reviewed drafts of the paper, patient samples preparation.
- Tamotsu Kanazawa analyzed the data, reviewed drafts of the paper.
- Yasuhiro Yoshida conceived and designed the experiments, performed the experiments, analyzed the data, wrote the paper, reviewed drafts of the paper.

## Human Ethics

The following information was supplied relating to ethical approvals (i.e., approving body and any reference numbers):

The committee of University of Occupational and Environmental Health, Japan, H23-061.

## Data Deposition

The raw data has been supplied as a Supplemental Dataset Files.

## Supplemental Information

Supplemental information for this article can be found online at http://dx.doi.org/10.7717/peerj.2026#supplemental-information.

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
