# Peer review of "Relationship between triterpenoid anticancer drug resistance, autophagy, and caspase-1 in adult T-cell leukemia"

_PeerJ, doi:10.7717/peerj.2026_

## Round 0.1 · original submission · Minor Revisions

Overall, the results are quite well described and support the conclusions drawn. However, both Reviewers provided comments and suggestions to improve the paper. In particular, the detailed criticism raised by Referee 1 should be carefully addressed before the paper can be published.

Reviewer 1 ·

Basic reporting

The manuscript “Relationship between triterpenoid anticancer drug resistance, autophagy, and capase-1 in adult T-cell leukemia” by Nakanishi et al. demonstrated the antitumorigenic effects of CuD and of the Bcl-xL inhibitor Z36, in primary cells from adult T cell leukemia (ATL) patients with respect to autophagy involvement and drug resistance, mediated by upregulation of caspase-1 expression. The study addresses a relevant question and presents interesting results in peripheral blood lymphocytes (PBL) isolated from ATL patients. The article is written using clear and unambiguous text, but there are some comments:
- In the Introduction section of the paper, the authors might introduce a reference when they referred to the development of drug resistance mechanisms to glucocorticoids treatment (line 46). Furthermore, they should improve the reported references with more current and relevant citations to describe the role of autophagy in treatment of T cell leukemia (line 60).
- In agreement with PeerJ manuscript organization, the authors should introduce in the paper the “Conclusions” section.
Figures reported in the study are relevant, but not all are well labeled and described. In particular:
- Fig 1: Panel D, the title of the vertical axis lacks in the graph (relative cell viability)
- Fig 3: Panel A, in the legend the authors should indicate the abbreviation ET and DM.
- Fig4: the author should invert Panel B with Panel C, because they in the text (Results section) discuss panel C before panel B.
- Supplementary Figure 2: Indicate Panel A (T cell lymphoma patients) and Panel B (healthy donors), in the figure.

Experimental design

The present study addresses a research questions within scope of the journal. The research question is well defined but there are some comments:
- Methods relative to the Preparation of human peripheral blood lymphocytes (PBLs) should be described with more information and the authors should introduce a reference in the text (line 89).
- In Materials and Methods section line 99, the authors should indicate the acronym RIPA (radioimmunoprecipitation assay lysis buffer).
- When the authors show the inhibition effect on proliferation induced by Z36 in ATL Patient 1 (Fig. 2B), it should be clearly indicated that this effect has been evaluated before chemotherapy. Furthermore, the authors should also introduce a graph of cell viability that represents a data pool obtained in ATL patient cells at the indicated concentrations for 24 h.
- When the authors show the results on prednisolone-treated ATL patient cells (Fig 2D), it is not indicated any characteristics about enrolled patients. This information could be introduced in the table 1.
- The results showed in the Fig. 3, relative to the cellular autophagy, are not clearly described in the text. In particular, the authors should introduce some comments about quantitative analysis of the expression levels of target proteins with or without treatment of cells with 3-MA.
- When the authors describe the Fig 4D in Results section, they should indicated, also in the text, that the data showed are referred to ATL patient cells obtained after umbilical cord blood transplantation and upon recurrence.

Validity of the findings

- The findings reported on ATL patient cells could have interesting clinical implication even if data relative to the occurrence of drug resistance after chemotherapy and after umbilical cord blood transplantation are referred only to 1 patient.
- In the Discussion section, the authors not address and explain very well the question related to the relationship between triterpenoid anticancer drug and cellular autophagy. The results showed in the Fig 1D and Fig 3A do not support the conclusion described in the discussion section, when the authors reported that the effects of the inhibitor CuD were enhanced in the presence of 3-MA (line183 and 184). Furthermore, the authors should introduce a more recent reference when they describe the role of autophagy in cancer and they should introduce a new reference in the text when they describe Z36 as an autophagy inducer in HeLa cells (line 188).

Reviewer 2 ·

Basic reporting

The last sentence of the Abstract should be rewritten highlighting the results obtained by CuD and Z36 on primary ATL blasts; it is instead focused on caspase1 upregulation in resistance.
English should be revised.
In the Materials and Methods and Results sections authors shoud point out that results on Z36 and CuD were confirmed in eight patients and three healthy controls and that only a representative case is shown. Experimental results are sometimes described as derived from a single patient and sometimes as derived from patients; it is confusing.
Legends to Figures 2 are 3 poorly written; for example, the meaning of ET and DM abbreviations should be indicated in the text.

Experimental design

No comments

Validity of the findings

No comments

Additional comments

The authors describe the death-inducing effect of the inflammosome inhibitor CuD on ATL blasts, also showing their increased sensitivity, as compared to normal PBLs. They demonstrate that CuD and Z36, a bcl-2 inhibitor, are synergic whereas autophagy negatively regulates CuD and Z36 activity. CuD activity decreases after ATL recurrence and this observation is linked to caspase 1 up-regulation.
The manuscript is presented in an intelligible fashion and current literature is reviewed. The experimental workup is described, it is technically sound and supports the conclusions. Therefore, I recommend the acceptance of the manuscript.

---

## Round 0.2 · accepted · Accept

The authors have addressed the comments and suggestions provided in the previous review. Therefore, the paper is now acceptable for publication in PeerJ.

Reviewer 1 ·

Basic reporting

No Comments

Experimental design

No Comments

Validity of the findings

No Comments

Additional comments

The authors have addressed the comments and suggestions provided in the previous review. There are no further comments